# Advanced Hand Gesture Prediction Robust to Electrode Shift with an Arbitrary Angle

**DOI:** 10.3390/s20041113

**Published:** 2020-02-18

**Authors:** Zhenjin Xu, Linyong Shen, Jinwu Qian, Zhen Zhang

**Affiliations:** School of Mechatronic Engineering and Automation, Shanghai University, Shanghai 200444, China; zhenjinxu@shu.edu.cn (Z.X.); shenlyshu@shu.edu.cn (L.S.); jwqian@mail.shu.edu.cn (J.Q.)

**Keywords:** simplified rapid correction, electrode shift, hand gesture prediction, sEMG, synchronous gesture

## Abstract

Recent advances in myoelectric controlled techniques have made the surface electromyogram (sEMG)-based sensing armband a promising candidate for acquiring bioelectric signals in a simple and convenient way. However, inevitable electrode shift as a non-negligible defect commonly causes a trained classifier requiring continuous recalibrations. In this study, a novel hand gesture prediction is firstly proposed; it is robust to electrode shift with arbitrary angle. Unlike real-time recognition which outputs target gestures only after the termination of hand motions, our proposed advanced prediction can provide the same results, even before the completion of signal collection. Moreover, by combining interpolated peak location and preset synchronous gesture, the developed simplified rapid electrode shift detection and correction at random rather than previous fixed angles are realized. Experimental results demonstrate that it is possible to achieve both electrode shift detection with high precision and gesture prediction with high accuracy. This study provides a new insight into electrode shift robustness which brings gesture prediction a step closer to practical applications.

## 1. Introduction

Hand gesture serves not only as an auxiliary enhancer of reinforcing information delivery in human conversations, but also as a primary method for transferring instructions with human–computer interaction devices [1]. The capability of machines to recognize distinctive gesture characteristics can be harnessed in a wide variety of applications [2,3]. The fast growing characteristic of surface electromyogram (sEMG) has made it a promising candidate for hand motion detection, recognition or even prediction (as bio-signal sensing technique is). While the sEMG signal cannot be solely utilized in discriminating dynamic or spatial hand gestures [4], it has particular advantages of non-invasive sensing and decoding fine muscular activity simply and directly. Additionally, compared with vision-based approaches which could be greatly affected by background illumination and complexity or other external factors [5], the sEMG signal can also provide a responsive way which is capable of eliminating interference by ambient light and noise to take full control of a prosthesis [6].

By attaching certain sets of electrodes to the skin, sEMG signals are typically recorded via electrical activities such as muscle contraction. Different hand gestures activate specific muscular regions; thus, hand or fingers movement intentions can be identified while performing multiple ones [7]. However, there are usually classification errors due to an electrode shift and when the wearing position may deviate from that of previous use [8]. Recent advances in wearable sEMG sensors (e.g., sEMG armband) have facilitated the process of bioelectrical signal acquisition, whereas electrode position should be consistent in case of reducing the classification accuracy [9]. The position-dependent properties of such bioelectrical signals are the critical challenge to make recognition performance less interference with subtle or larger position change during practical use.

The conventional solutions to address the above classification issues can be categorized into two basic strategies: training the classifier for general or individual [10]. General recognition classifiers are trained with a standard and open-access but sample-limited dataset which requires implementing the same electrode configuration to allow data collection regularization. To further train and test the general recognition systems, the sensing armband ought to be placed on the exactly unaltered position which seems unlikely to achieve in actual applications [11]. In comparison to general classification models, although training an individual classifier does not need to have a pre-defined consistent wearing style, its model has to be retrained and retested on every new session [12]. Take into account user experience; a previous case study implies that amputees who have to refresh prothesis up 3.2 times per day are willing to recalibrate the classifier no more than every 2.4 h [13]. Therefore, selecting an optimal training strategy and minimizing calibration times are the key concerns which should be addressed in priority.

However, there has been little systematic literature that focus on exploring the effective solutions to ring-armband position-changed issues. A previous study published by Zhang et al. [14] indicates that sensor rotation strongly deteriorates the accuracy of classification model training by signals acquired from one position. Specifically, the proposed model can only make rotating correction and remapping of every 45°. Li et al. [15] initially established a polar coordinate to measure shifting angles, whereas predicted rotating resolution limits to ± 45° as well. Steinhardt et al. [16] take advantage of particular time-domain features (e.g., mean absolute value) to repeatedly characterize orientation shift between initial and new wearing position. Vimos et al. [10] adopted one training set that was acquired from the same suggested position; the involved model can only present higher accuracies on π/4-based positions.

Furthermore, to our knowledge, many state-of-the-art research regarding real-time improvement of hand gesture recognition still only pay attention to investigate the critical parameters which may cause human perceivable delay [17]; for example, feature [18,19] and classifier selection [20,21,22,23,24]. However, their common trait is that the following data processing procedures have to wait until sEMG recording is finished (i.e., motions ended). Although recognition technologies present good real-time performance [25,26], there is an inherent flaw–procedure separation of signal acquisition and processing as indicated above. So, the method of conducting data preprocessing and extracting features while bioelectrical signals are being collected is a novel point to gain a decent real-time response, endowing the model with the capacity of gesture prediction.

In this paper, we firstly report a hand gesture prediction model with an emphasis on estimating electrode shift effects to provide a new insight into wearing-independence based on an sEMG armband. The main contributions of our work are as follows:the proposed model enables advanced predictive capability via an improved artificial neural network (ANN) substantially, which could output predicted results in 338 ms from hand gestures start, with above 94% accuracy;the developed method allows electrode displacement detection at random angles rather than conventional fixed coarse resolution, capable of satisfying actual requirements;this system can make simplified rapid correction according to electrode shift.

As a demonstration of our proposed solution’s utility, we randomly conduct eight independent experiment sessions over time in which the subjects could perform various wearing styles. The method of effectiveness and robustness to ring–electrode rotation, together with a predictive classifier, show promising potential in making hand gesture recognition or even prediction more applicable in practice.

We describe the materials and methods in Section 2; in Section 3, the experimental results demonstrate the decent performance both of electrode shift detection and improved ANN-based hand gesture prediction. Finally, we will discuss the novelty of our proposed prediction model and shift correction strategy, then give a brief conclusion.

## 2. Materials and Methods

There are two primary objectives of this work: 1. To investigate the optimal strategy for correcting electrode rotation shift with an arbitrary angle; 2. To realize hand gesture prediction. By means of a wearable armband mounted around the forearm, we are able to access raw sEMG data from either initial suggested or new shifting positions. Then, the raw acquired signal data can be preprocessed, including data normalization; low-pass filtering for removing sEMG noise is used to make the collected information more representative of target gestures. 

Meanwhile, for reducing electrode shift, the proposed method underlying robustness to the position-shifted model contains four basic procedures, including electrode shift identification by interpolated peak location (IPL), signal matrix rearrangement, three-standard shifting training and multi-session prediction testing, as shown in Figure 1. It should be noted that the trained model combining majority voting needs to output target gestures before motion is finished. The involved sEMG sensing device, developed rotating correction and hand motion prediction method will be described in detail as follows.

### 2.1. Electrode Registration and Data Acquisition

All experiments are conducted with ten able-bodied subjects (two females included, 24 ± 1.5 years) by Myo armband, a commercially available sEMG sensor, which consists of eight circular-distributed bipolar dry electrodes. This sensor is capable of normally measuring forearm muscular electrical activity with a sampling frequency of 200 Hz. Additionally, the collected sEMG data can be transmitted to host computer (OS: Windows 10; CPU: i7-9750H; RAM: 16 GB) via built-in Bluetooth in real time; then, with a front panel graphical user interface for data display and via MATLAB for next data processing and analysis. 

For data preprocessing and segmentation, we implement an initial data process on the smoothed signals with marking the region of muscular activity. This practice is achieved by calculating the energy spectrum using short-time Fourier transform (STFT), which can eliminate inactive intervals and keep effective information. Then, we employ sliding windows to convert signal data into feature vectors. The window length in this study is 200 ms with a step of 5 ms, and every sliding window can automatically align with the onset of the muscular activity region and slide along the processing direction.

#### 2.1.1. Standard Configuration and Initial Position Designation

The armband manufacturer recommends a specific wearing style as standard configuration; the logo-printed channel (channel 4 or CH_4_) should be approximately aligned with subject’s middle finger, as illustrated in Figure 2a. To avoid more variables while acquiring signals and make contrast with previous studies, we designate the above-suggested configuration as the initial position for the next rotation shift analysis.

#### 2.1.2. Employed and Synchronous Gesture Definition

Six typical hand gestures (Figure 2b) are employed with the Myo armband, including hand closed (HC), hand opened (HO), wrist extension (WE), wrist flexion (WF), double tap (DT) and no movement (NM). Subjects were asked to finish each gesture in 2 s. Specially, we also define a synchronous gesture (WE) to predict the rotation angle at onset of each independent session, and the above synchronous operation is merely required to be performed once.

#### 2.1.3. Training and Testing Dataset Organization

To realize correcting rotation shift with arbitrary angle and build a user-specific model using minimum training data, we took three representative positions to perform these selected gestures. It should be noted that every single gesture (gesture NM included) only comprises five repetitions. To evaluate the trained ANN classifier, subjects were required to repeat each gesture (gesture NM excluded) 15 times in one independent session regardless of wearing style; the testing set consists of eight sessions over time. For every repetition, the subjects started with his/her arm relaxed, and then performed designated gestures, then returned to the initial relax position. Notably, before performance evaluation, sEMG data acquirement from three training wearing positions was conducted independently.

### 2.2. Electrode Shift Detection Based on IPL and Synchronous Gesture

Although sEMG signals can be recorded in a simple-operated way, the properties of non-linearity and non-stationarity are considered to be barriers to the achievement of high classification accuracy [27]. Inspired by feature-based methods, we attempted to take advantage of optimal feature to quantitatively measure electrode shift during session switching. That is, the shifted electrode position can be corrected to standard configuration only when the shift angles are confirmed, then feed the proofed data to a trained ANN classifier to make gesture predictions. To address this problem, we propose a specific gesture based on a signal synchronization method [28] that can activate minimum muscular regions as muscle contraction. This work uses IPL to detect the maximum interpolated amplitude via the sum of preprocessed data in each sensing channel, and then the interpolated curve can be plotted using a cubic spline interpolation as shown in Figure 3a.

Concerning the proposed rotation shifting prediction strategy with high accuracy of arbitrary angle, there is a need to increase the position-correction tolerance of the present model. Therefore, Standard Space is defined to regularize, and then we make a rearrangement of the raw signals allowing the position detection model to be robust to sensor shift. After electrode shifting status is assured, the raw channels will be transformed to Standard Space, which are ordered by the highest sum channel using IPL. Moreover, we choose WE to serve as the synchronous gesture since both maximum signal value and data sum consistently concentrate in one specific channel. Experimentally, the tolerance of Standard Space is increased to ± CH_4_ (Figure 3b). To further identify the specific offset orientation with arbitrary angle, we apply the maximum data value of interpolated amplitude to represent electrode shift.

We also provide a schematic view (Figure 3a) to illustrate that the synchronous process can be activated by gesture WE, as shown in the blue-dotted zone. Meanwhile, the shift-corrected angle at a new position can be expressed precisely as 130° rather than the initial CH_4_ angle (135°). Specifically, raw sEMG signals acquired from the sEMG sensor are represented as [29]:(1)$$I\left(n\right)={\left(I_{1}\left(n\right),\ldots ,I_{8}\left(n\right)\right)}^{T}$$

Each employed gesture takes 2 s with the Myo armband which normally works with a sampling rate of 200 Hz, that is, it contains 400 items of minimum measurement unit **I***(n)*. It is defined as:(2)Sn=(I(n−399),…,I(n))T∈ℝ400×8

Moreover, signal preprocessing can be expressed as:(3)$$F=\Psi [\mathrm{abs}[S_{n}]{]}^{400\times 8}$$
where **S**_n_ is raw sEMG data acquired in 2 s, **Ψ** represents a 4th order low-pass Butterworth filter with a cutoff frequency of 5 Hz. We can add the items in each column to seek the highest sum of signal channel data (N*_CH4_*) which is defined as:(4)$$N_{CH4}=\max{\left(\sum^{\;}F\left(1\right),\ldots ,\sum^{\;}F\left(8\right)\right)}^{1\times 8}$$

### 2.3. Data Rearrangement

By means of the above electrode shift detection using IPL, the armband rotating position can be confirmed and transformed to Standard Space. Therefore, the shifted degree can be expressed via the offset angle whose channel possesses maximum interpolated amplitude relative to initial electrode configuration. To correct the interfered sEMG samples, signal data have to be rearranged to Standard Space. The correction procedure is only done once at the onset of each experiment session by performing synchronous gestures. The rearrangement method can be briefly described by establishing a polar coordinate as illustrated in Figure 4, and all the sampling points which are collected in one gesture duration (2 s) are present in eight polar axes in steps of π/4, respectively. Then, the identified peak signal polar can be rotated counterclockwise to 3π/4, and the shifted channel satisfies the definition of Standard Space.

Applying the above rearrangement strategy, shifted position **P**’ can be transformed to Standard Space by raw position **P** as calculated in the following:(5)$$P_{\left(mod\left(k+n-4,\;\;8\right)\right)}\to P^{\prime },\;n=1,2,\ldots ,8$$

Similarly, shifted sEMG data **I**’*(n)* of each electrode channel can be expressed as:(6)$$I\left(mod\left(k+n-4,\;8\right)\right)=I^{\prime }\left(n\right),\;n=1,2,\ldots ,8$$
where *k* is the channel of maximum interpolated amplitude, *mod* means the remainder after division. Supposing *k* is 2 (instance as Figure 4), the relation between raw and shifted channel is listed in Table 1.

### 2.4. Training Set Selection

The above shift detection and correction allow the raw shifted data to transform to the unified sEMG data arrangement on Standard Space. By means of these data, we can train one classifier without taking random shift conditions into consideration. To evaluate the shift-corrected performance at various sessions over time, we employ the above strategy of shift detection and data rearrangement, and set the criterion of whether the corrected position presents under the tolerance of Standard Space (i.e., ± CH_4_ or 135° ± 22.5°). As shown in Figure 5a, the corrected rotation offsets among all testing sessions are confined to Standard Space. Meanwhile, we also select three representative wearing positions (close to 135° and 135 ± 22.5°) to organize the training set (Figure 5b).

### 2.5. ANN-Based Hand Gesture Prediction

Five time-domain features are selected for further extracting characteristic information from preprocessed data, including mean absolute value, root mean square, slope sign change, wave length, and Hjorth parameter. For outputting predicted gestures before motions are finished and decreasing spurious misclassification errors, we add postprocessing to the last link of prediction system. By combining the majority voting method [30], every sliding window (200 ms or 40 sampling points) generates one specific code in accordance with predefined gestures via the prediction function of educated classifier. Then, the current window shifts into the next with the step of 5 ms. If the gestural labels accumulate to the threshold (55), the postprocessing terminates. 

Our forward-propagation ANN model is implemented with three layers and trained by using full batch gradient descent with cross-entropy cost function, and applied regularization using weight decay (λ = 150). The number of input layer nodes is 376, that is, the product of one sliding window length and total channel number (40 × 8 = 320), plus features extraction functions creating a vector of 56 features (7 × 8 = 56), corresponding to the length of feature vectors. Meanwhile, we take the size of a hidden layer to be half of the input one using the *tanh* transfer function [12]. Moreover, we set six nodes on the output layer in order to correspond to the six pre-defined gestures (including NM). The specific prediction procedure will be detailed at the Algorithm 1. **Algorithm 1.** ANN-based prediction model.**Input:**$F_{\left(n\right)}^{\prime },\;T_{\left(n\right)}\leftarrow corrected\;sEMG\;\&\;testing\;set$**Output:**1, 2, 3, 4, 5 ←gesture code1:initialize *s*; // *where s denotes stride length*2:initialize ξ(X); *// where ξ(X) denotes time-domain features bag function;*3:training process: extract features from $F_{\left(n\right)}^{\prime }$ using  ξ(X) to form $\left(F_{i},\;L_{i}\right)\leftarrow \xi \left(F_{\left(n\right)}^{\prime }\right)$
 *// where F_i_ and L_i_ denote feature vector and corresponding label vector, respectively*4:compute weight decay && employ regularization to F←(Fi−μ)σ
5:apply (F, Υ) to *classifier*, and form *classifier.predict*6:for each windowed $T_{\left(n\right)}$, feed $\xi \left(T_{\left(n\right)}\right)$ to *classifier* for predicting *gesture code*7:count number of different generated *code* to form *n*8:if *n* == preset *threshold* of outputting gesture, then 
  get the predicted *gesture code*, else return *None*

## 3. Experimental Results

### 3.1. Prediction Accuracy with Electrode Shift Correction

Applying the above electrode shift correction strategy, Figure 6a shows the overall classification accuracy for all gestures with various sessions. For each group of presented hand gestures, the corresponding prediction results are much better than the control groups in Section 3.2. The highest and lowest precisions for hand motions are WF (98.6%) and HO (88.8%), respectively. With respect to sensitivity, the gesture HO has the highest rate (96.2%) and WF has the lowest (93.1%). Furthermore, Figure 6b highlights the developed prediction model for the responsive superiority with an averaged response time 338 ms. It should be noted that this prediction procedure starts at the onset of hand motion, and terminates at the corresponding response time. Therefore, the prediction results can be presented before gestures are completed, eliminating external device delay as perceived by users [31]. 

### 3.2. Accuracy Improvement on Electrode Shift

In this study, we also explore the governing parameters that may significantly influence the prediction accuracy, including the number of wearing positions during training data acquisition. The gesture prediction accuracy comparison between electrode shift corrected and non-corrected situations is summarized as shown in Table 2. Specifically, as for non-correction of electrode shift, the lowest and highest accuracy improvements are SUB #3 (12.0%) and SUB #04 (71.7%), respectively. This is probably because of distinctive wearing styles and forearm girths among individuals.

Moreover, the OVERALL row of Table 2 indicates that the system performance will decrease dramatically (fall from 94.7% to 51.4%) without proposed correction strategy. The averaged prediction results with one specific wearing style also shows an accuracy of 72.0% (also a drop from 94.7%), which is not up to the standard of real applications.

### 3.3. Synchronous Gesture Selection Varies Accuracies

A comparison is also made in this work among different gestures to emphasize the significance of selecting the appropriate synchronous gesture. From Table 3, a certain prediction accuracy of lifting varies among subjects and could be easily realized while shifting from gesture WF to WE; the overall accuracy is improved by 34.2%.

## 4. Discussion

With the considerable advancements and a broad range of applications of sEMG-based technologies, myoelectric-controlled terminals (e.g., prostheses) have pointed out more greater requirements for real-time system responses, simplified rapid error-correction and high recognition or even prediction accuracy. Specifically, in clinical scenarios, it is inevitable that the electrode shift happens during the sEMG sensors being taken on and off, which will definitely cause degradation of classification performance [32]. Thus, this study reveals that the electrode displacement correction of an arbitrary angle is a crucial necessity to enhance robustness on hand gesture prediction.

### 4.1. Governing Parameters Varies Accuracies

In this research, we analyze the governing parameters that may cause degradation of prediction performance without correcting electrode shift and selecting enough training sets or appropriate synchronous gestures. From the analysis of the given clues, it is more obvious that the classification accuracy for all gestures among different experimental sessions are all improved to varying degrees.

Meanwhile, as for selecting an appropriate number of training wearing positions, solely taking advantage of one preset electrode arrangement whose electrode position is almost or exactly the same as the standard configuration presents lower accuracy than choosing three positions. That is, there is an accuracy lifting that appropriately chooses more training positions which can enable prediction model to better cope with the electrode shift with arbitrary angles.

Obviously, as Figure 7 illustrates, there are two activation muscular regions while performing WF, whereas WE only presents one activation region which could minimize interference in shift detection. Notably, the signals from non-boxed channels can be filtered at preprocessing; however, other hand motions involved in this paper always possess rather complex sEMG characteristics which is hard to be filtered effectively.

### 4.2. Performance of Electrode Shift Correction

There is a large volume of literature published regarding non-shift occurrences, and various classification methods and data processing strategies have been applied on gesture recognition with decent performance. However, electrode shift should be considered together with practical use, and this factor influencing accuracies have been shown by less studies as listed in Table 4. As Table 4 shows, the sEMG armband and high-density electrodes are the primary sources to obtain raw acquired bioelectrical signals. Note that resolutions of electrode displacement correction in previous studies range roughly between 22.5° to 45°, or 1 cm or 2 cm shift, thereby showing much resistance towards clinical or practical deployments. 

Meanwhile, the response time (i.e., real-time performance evaluation) of recognition systems has not been investigated via related algorithms, reducing the introduction of gesture prediction techniques. The proposed strategy for correcting electrode displacement endows the model with the ability of shift detection with any arbitrary angle. Various experiments also demonstrate that the preset hand motions could be accurately identified, no matter how electrode rotation is offset away from the standard configuration. The main breakthrough also presents gesture prediction which evidently achieves high real-time classification accuracy.

### 4.3. Performance with Gesture Prediction

The final experimental result is related to the evaluation of predictive accomplishment via our improved ANN-based gesture prediction model. The postprocessing using majority voting compensates for the device intrinsic delay, which highlights the superiority of the prediction strategy. In the specific classification case of postprocessing, the trained classifier will generate one gesture code or label corresponding to one certain target hand motion in the same step of sliding window processing, then the output labels are added up to calculate the sum of each emerged gesture code. Eventually, the present model determines the final predicted gesture depending on which cumulative result reaches the threshold first. The scheme implementation superiority of majority voting for predicting rather than recognizing is not needed to obtain the entire part of the sEMG signals, which provides proof of the high real-time performance based on our improved data processing method.

Besides, compared with the strategy of conventional training and testing dataset segmentation, the current work focuses on implementing the proposed prediction model plus electrode shift detection in real scenarios. We believe that this novel strategy could provide a proof-of-concept for an innovation in organizing acquired bioelectrical data, to better stimulating practical application and evaluating classification performance.

## 5. Conclusions

In this research, we firstly propose a novel strategy for enhancing robustness to electrode displacement with any arbitrary angle on hand gesture prediction. Compared with traditional shift-correction methods with fixed coarse resolutions, the developed method allows electrode displacement prediction to be performed at random angles via IPL and synchronous gestures. In addition, our improved ANN-based model has dramatically improved hand gesture prediction by combining majority voting. This study also provides a new insight into a simplified rapid correction strategy on electrode shift with the promising potential in practical applications.

## Figures and Tables

**Figure 1 sensors-20-01113-f001:**
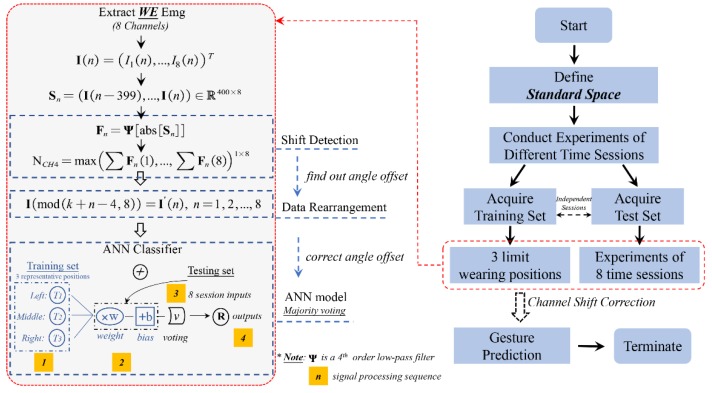
Scheme of adaptive channel shift strategy using preprocessed data plus improved ANN classifier with majority voting method. The electrode shifting position can be identified only using the predefined synchronous gesture, and once the changed configuration is established during acquiring training data from three representative wearing positions, the subject does not require further recalibrate the algorithm on one session.

**Figure 2 sensors-20-01113-f002:**
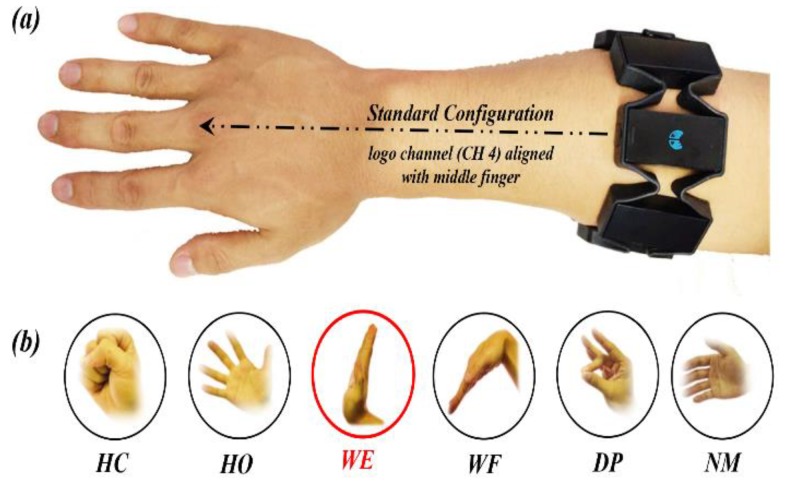
Scheme of standard electrode configuration and applied hand gestures. (**a**) Logo channel, also defined as CH_4_ is suggested to be aligned with middle finger. (**b**) Red-marked synchronous gesture, WE is used to identify electrode shift. Here hand closed (HC), hand opened (HO), wrist extension (WE), wrist flexion (WF), double tap (DT) and no movement (NM).

**Figure 3 sensors-20-01113-f003:**
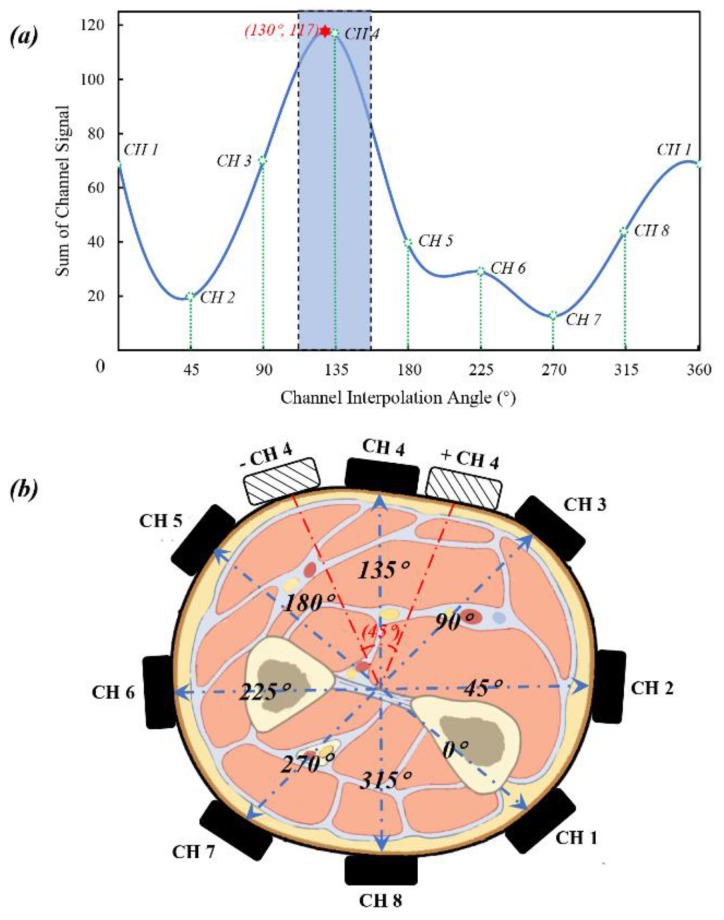
Scheme of initial position on right forearm and IPL by WE. (**a**) Based on sum of each channel signal, the interpolated curve can be plotted using a cubic spline interpolation, and the shifting angle can be represented as red-marked point (130°). (**b**) One full electrode position range is divided into 45°, and the tolerance of Standard Space is increased to ± CH_4_ as red-dotted zone.

**Figure 4 sensors-20-01113-f004:**
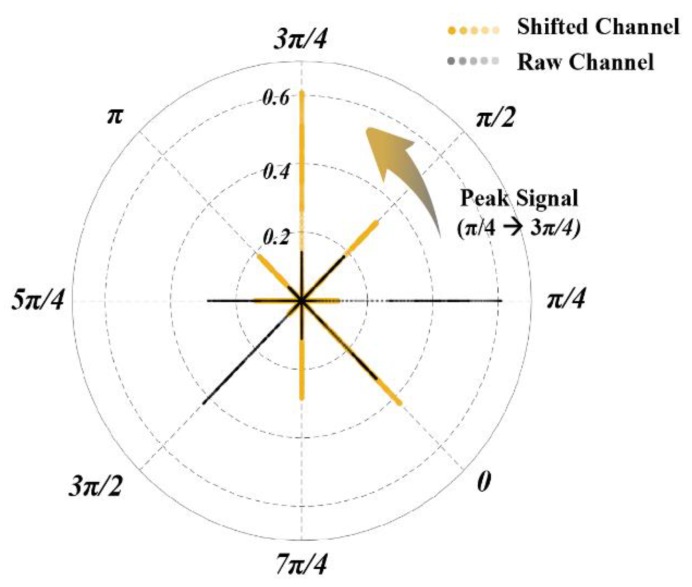
Polar scatter of every sampling point in each sensing channel on 8 polar axes. Raw channel is transformed to shifted channel according to maximum channel amplitude.

**Figure 5 sensors-20-01113-f005:**
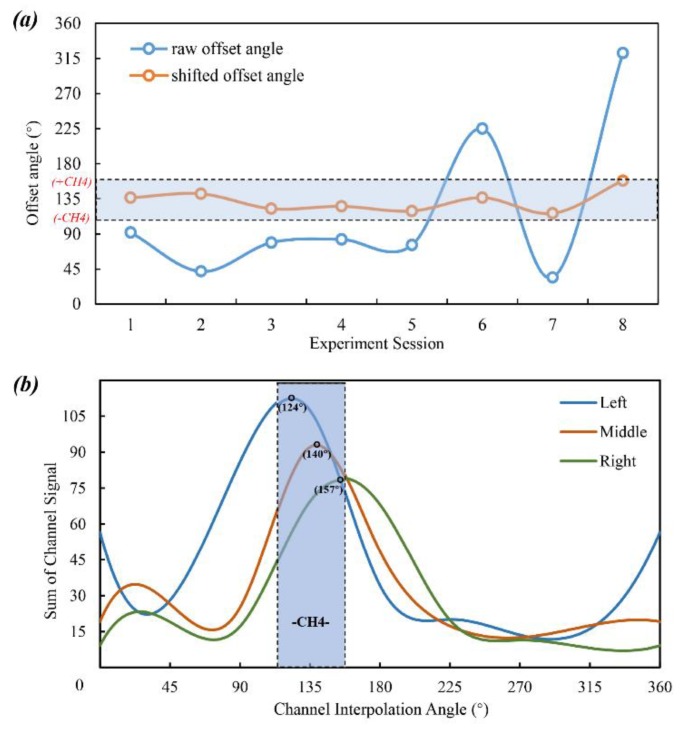
(**a**) Electrode shift on various experiments. (**b**) 3 representative wearing position for acquiring training set.

**Figure 6 sensors-20-01113-f006:**
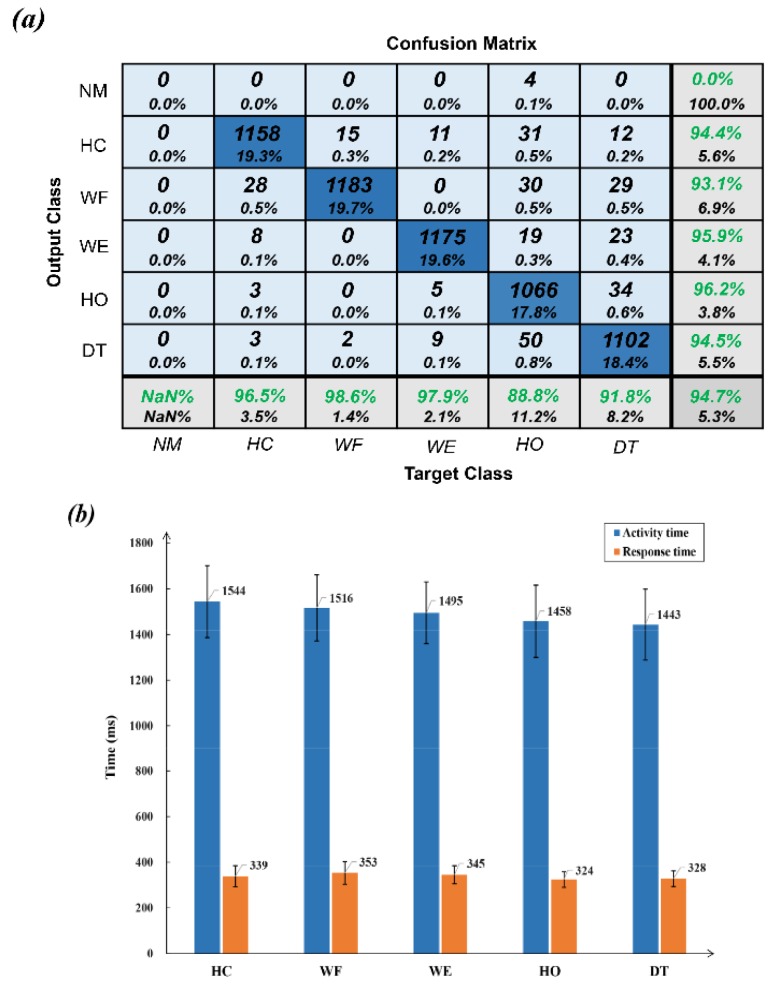
(**a**) Confusion matrix for the proposed hand gesture prediction model with electrode shift correction. (**b**) Comparison of averaged prediction response and activity time on each target hand gesture.

**Figure 7 sensors-20-01113-f007:**
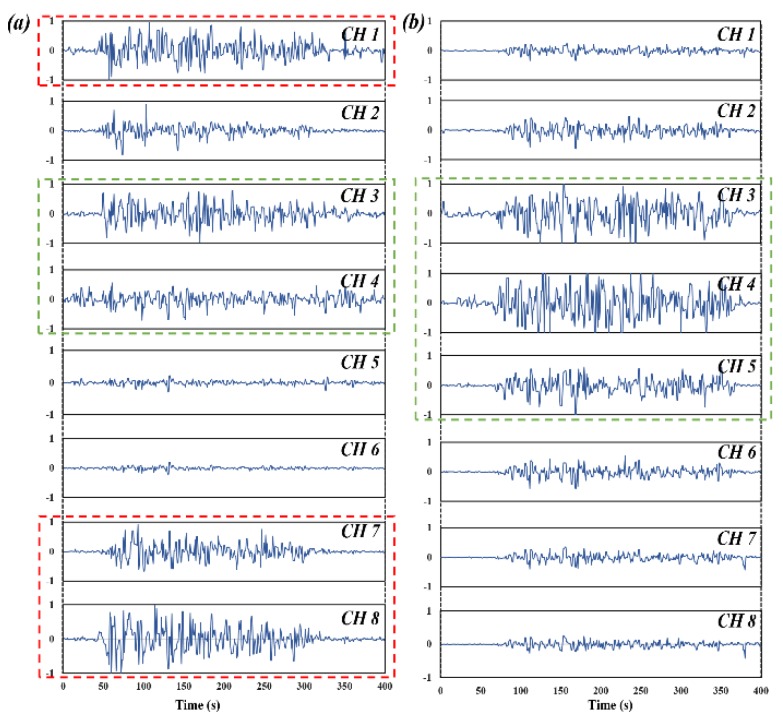
Acquired raw sEMG signals of two pre-defined gestures from Myo armband mounted on forearm. (**a**) WF. (**b**) WE. Red dashed box represents anterior forearm muscular region, green dashed box represents posterior forearm muscular region.

**Table 1 sensors-20-01113-t001:** The relation between raw and shifted channel.

Electrode Position	sEMG Rearrangement
2 → 4	I(2) = I’(4)
3 →5	I(3) = I’(5)
4 → 6	I(4) = I’(6)
5 → 7	I(5) = I’(7)
6 → 8	I(6) = I’(8)
7 → 1	I(7) = I’(1)
8 → 2	I(8) = I’(2)
1 → 3	I(1) = I’(3)

**Table 2 sensors-20-01113-t002:** The classification accuracy for all gestures with different experimental sessions.

	M_1_	M_2_	M_3_	C_1-3_	C_2-3_
SUB #01	73.7	55.3	94.0	20.3	38.7
SUB #02	69.8	78.3	94.0	24.2	15.7
SUB #03	79.3	72.0	91.3	12.0	19.3
SUB #04	21.3	60.0	93.0	71.7	33.0
SUB #05	60.8	59.5	96.2	35.4	36.7
SUB #06	28.2	89.0	97.5	69.3	8.5
SUB #07	48.5	63.2	90.7	42.2	27.5
SUB #08	34.2	81.3	95.7	61.5	14.4
SUB #09	36.5	78.2	96.8	60.3	18.6
SUB #10	61.5	83.5	98.2	36.7	14.7
OVERALL	51.4	72.0	94.7	43.3	22.7

M_1_ represents the method without using proposed electrode shift correction strategy; M_2_ stands for the method merely using 1 wearing position to acquire training dataset; M_3_ represents our proposed shift-correction strategy based on improved ANN; C_1-3_ stands for prediction accuracy lifting compared M_3_ with M_1_; C_2-3_ stands for prediction accuracy lifting compared M_3_ with M_2_. Unit: %.

**Table 3 sensors-20-01113-t003:** The classification accuracy for all gestures with different synchronous gestures.

	S_1_	S_2_	C_1-3_
SUB #01	42.7	94.0	51.3
SUB #02	61.7	94.0	32.3
SUB #03	80.5	91.3	10.8
SUB #04	29.7	93.0	63.3
SUB #05	33.0	96.2	63.2
SUB #06	29.8	97.5	67.7
SUB #07	71.3	90.7	19.4
SUB #08	84.7	95.7	11.0
SUB #09	72.5	96.8	24.3
SUB #10	98.7	98.2	−0.5
OVERALL	60.5	94.7	34.2

S_1_ represents the shift correction method by WF; S_2_ stands for the method by WE; C_1-3_ stands for prediction accuracy lifting compared S_2_ with S_1_. Unit: %.

**Table 4 sensors-20-01113-t004:** The results of different hand gesture recognitions.

Task	Work	Electrode	Channel	Classifier	Gesture	Response Time	Resolution of Shift Correction	Accuracy
**Advanced prediction**	**OUR**	**dry**	**8**	**ANN**	**6^*^**	**338ms (< GD)**	**arbitrary angle**	**94.7%**
*Traditional recognition*	Li et al. [15]	dry	8	SVM	8^*^	GD^+^	fixed (45°)	78.4%
	Vimos et al. [10]	dry	8	SVM	6^*^	GD^+^	fixed (45°)	92.4%
	Steinhardt et al. [16]	dry	8	SRC	6^*^	GD^+^	fixed (22.5°)	95.7%
	Zhang et al. [14]	dry	8	RF	15	GD^+^	fixed (45°)	91.5%
	Lv et al. [33]	SA	192	SAE	10^*^	GD^+^	fixed (1-cm shift)	85.0%
	Fan et al. [34]	SA	30	LDA	11^*^	GD^+^	fixed (1-cm shift)	88.2%
	Yang et al. [35]	dry	8	CNN	10^*^	GD^+^	fixed (45°)	63.2%

* Note: SA (self-adhesive electrode); GD (gesture duration); n* represents total gesture number, including no-movement (NM).

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
