# Peer review of "Advanced Hand Gesture Prediction Robust to Electrode Shift with an Arbitrary Angle"

_sensors, 2020, doi:10.3390/s20041113_

Round 1

Reviewer 1 Report

The proposed prediction method is novel and interesting. However, the description of the method is somehow not logical enough, making it a bit unfriendly to understand.

The study proposed a prediction method, which can optimize the signals recorded by Myo armband, to recognize gesture under arbitrary wearing position. It can be summed that the prediction includes three fundamental parts:
(1) Using the IPL method, the NCH4 and k were inferred for data rearrangement to shift the offset angle located into the Standard Space.
(2) Additionally, the ANN was trained with data from three adjacent wearing positions which are all inside the Standard Space.
(3) To conduct the prediction, the preprocessed data was involved to validate the algorithm, which is combined using ANN and the majority vote method.

The following are some questions:
(1) Though the sEMG trace instrument is commercial, the process of how the saved data was utilized is not presented. For example, what's the sample-frequency for feature calculation as well as the software environment? Since the prediction is ANN-based, then where and how the neural network was constructed? More necessary descriptions of the data modeling would be better.
(2) In the 8 testing experiment sessions, each hand gesture was repeated 15 times. But, in the confusion matrix, the gesture NM is not involved (NaN%) as shown in Figure 6(a) and 6(b). Is the sEMG data of NM not recorded for ANN training or testing? Please be more specific in data preparation either.
(3) It seems that several calculation methods were introduced into the prediction process, like the IPL, the majority vote, and the low-pass Butterfly filter. However, there is not enough information that helps explain why and how these methods were involved in the prediction.
(4) The prediction primarily consists of three key links (shift detection using IPL, data rearrangement to achieve to Standard Space, and ANN combining the voting method), while the context or logical sequence is not described obviously. The relationships between every two parts were not expressed well to make the method integrated. So it feels confused when understanding the comprehensive function of each link. Also, maybe the scheme in Figure 1 can be improved by adding the titles of the three links and mark out their relationships.
(5) It seems that Formula (2) misses a transposition. Please check it.
(6) Figure 3(b) was referenced in the context before Figure 3(a), which might be inappropriate in the order.
(7) The discussion section includes content that seems to be experimental results. Please distinguish the content of results and discussion, and list them in the proper section.
(8) Line 195 mentions session 5, which appears to be a small mistake.
(9) Some narration is repetitive. For instance, lines 109, 128 and 213 seem to all explain the same thing, that of 8 experiment sessions used for evaluating the prediction. And, in line 213, it says that the 8 independent experimental sessions were used to organize the training and testing datasets. Then, does this mean that the 3 experiment sessions for organizing the ANN training dataset were included in these 8 sessions? If so, is the training data used for testing? This kind of approach is unfair during ANN evaluation.

Author Response

GENERAL RESPONSE: We sincerely thank the editor and all reviewers for their valuable feedback that we have used to improve the quality of our manuscript. The reviewer comments are laid out below in italicized font and specific concerns have been numbered. Our response is given in normal font and changes/additions to the manuscript are using the "Track Changes" function in Microsoft Word.

POINT-TO-POINT RESPONSE:

COMMMENTS FROM REFEREE #1:

The proposed prediction method is novel and interesting. However, the description of the method is somehow not logical enough, making it a bit unfriendly to understand.

The study proposed a prediction method, which can optimize the signals recorded by Myo armband, to recognize gesture under arbitrary wearing position. It can be summed that the prediction includes three fundamental parts:

(1) Using the IPL method, the NCH4 and k were inferred for data rearrangement to shift the offset angle located into the Standard Space.

(2) Additionally, the ANN was trained with data from three adjacent wearing positions which are all inside the Standard Space.

(3) To conduct the prediction, the preprocessed data was involved to validate the algorithm, which is combined using ANN and the majority vote method.

The following are some questions:

Response: We appreciate the reviewer’s approval on our manuscript. According to your suggestions, we made several modifications to our previous draft, and the response to the pointed-out issues are as follows.

Though the sEMG trace instrument is commercial, the process of how the saved data was utilized is not presented. For example, what's the sample-frequency for feature calculation as well as the software environment? Since the prediction is ANN-based, then where and how the neural network was constructed? More necessary descriptions of the data modeling would be better.

Response: We thank the reviewer for raising this issue, and are sorry for not presenting further explanation on above-mentioned aspects. As such, we have provided the following necessary descriptions of data utilization (in section 2.1) and modeling (in section 2.5).

“All experiments are conducted with 10 able-bodied subjects (2 females included, 24 ± 1.5 years) by Myo armband, a commercially available sEMG sensor, which consists of 8 circular-distributed bipolar dry electrodes. This sensor is capable of normally measuring forearm muscular electrical activity with sampling frequency of 200Hz. Additionally, the collected sEMG data can be transmitted to host computer (OS: Windows 10; CPU: i7-9750H; RAM: 16GB) via built-in Bluetooth in real time, then, with a front panel graphical user interface for real time data display and via MATLAB for post data processing and analysis.” (in section 2.1)

“…By combining the majority voting method [30], every sliding window (200ms or 40 sampling points) generates one specific code in accordance with predefined gestures via the prediction function of educated classifier. Then, the current window shifts into the next with the step of 5ms. If the gestural labels accumulate to the threshold (55), the postprocessing terminates.

Our forward-propagation ANN model is implemented with three layers and trained this network by using full batch gradient descent with cross-entropy cost function, and applied regularization using weight decay (λ = 750 / total number of vectors). The number of input layer nodes is 376, that is, the product of one sliding window length and total channel number (40 * 8 = 320), plus features extraction functions creating a vector of 56 features (7 * 8 = 56), corresponding to the length of feature vectors. Meanwhile, we find out when the optimal size of hidden layer is half of input one using tanh transfer function, the classifier can get best prediction performance in such setting. Moreover, we set 6 nodes on the output layer in order to correspond to the 6 pre-defined gestures (including NM). The specific prediction procedure will be detailed at Algorithm.” (in section 2.5)

In the 8 testing experiment sessions, each hand gesture was repeated 15 times. But, in the confusion matrix, the gesture NM is not involved (NaN%) as shown in Figure 6(a) and 6(b). Is the sEMG data of NM not recorded for ANN training or testing? Please be more specific in data preparation either.

Response: Thank you for your careful work and thoughtful comment. In fact, we actually recorded sEMG data of NM for both model training and testing, but these NM datasets were only applied in training process, not in testing process. There are two main reasons for explaining such practice. Firstly, with NM data joining in training set, such move is capable of making the educated ANN classifier more accurate and adaptable to predict target gestures. Secondly, we have set a threshold for filtering low sEMG signal both in training and testing set, and this practice can effectively remove interference from NM data. Therefore, we manually removed NM data before classifier testing. Besides, we have also added the following details on data preparation as you suggested in section 2.1.3.

“It should be noted that every single gesture (gesture NM included) only comprises 5 repetitions. To evaluate the trained ANN classifier, subjects are required to repeat each gesture (gesture NM excluded) 15 times in one independent session regardless of wearing style, the testing set totally consists of 8 sessions over time.”

It seems that several calculation methods were introduced into the prediction process, like the IPL, the majority vote, and the low-pass Butterfly filter. However, there is not enough information that helps explain why and how these methods were involved in the prediction.

Response: The reviewer is right, and we have provided the following explanations to make more specific on presenting the proposed prediction method in section 2.

“By means of wearable armband mounted around forearm, we are able to access raw sEMG data that from either initial suggested or new shifting positions. Then, the raw acquired signal data can be preprocessed, including data normalization, low-pass filtering for removing sEMG noise, to make collected information more representative of target gestures.

Meanwhile, for reducing electrode shift, the proposed method underlying robustness to position-shifted model contains 4 basic procedures, including electrode shift identification by interpolated peak location (IPL), signal matrix rearrangement, three-standard shifting training and multi-session prediction testing, as showed in Figure 1. It is should be noted that the trained model needs to output target gestures before motions finished, and the proposed can achieve that with combing majority voting. The involved sEMG sensing device, developed rotating correction and hand motion prediction method will be described in detail as follows.”

The prediction primarily consists of three key links (shift detection using IPL, data rearrangement to achieve to Standard Space, and ANN combining the voting method), while the context or logical sequence is not described obviously. The relationships between every two parts were not expressed well to make the method integrated. So it feels confused when understanding the comprehensive function of each link. Also, maybe the scheme in Figure 1 can be improved by adding the titles of the three links and mark out their relationships.

Response: We greatly appreciate the reviewer for pointing this out. We have added the following sentences (in section 2.2) to make every part of gesture prediction more logical and consistent. Meanwhile, Figure 1 has been improved as you suggested. Thanks again.

“Inspired by feature-based method, we attempt to take advantage of optimal feature to quantitatively measure electrode shift during session switching. That is, the shifted electrode position can be corrected to standard configuration only when the shift angles confirmed, then feed the proofed to trained ANN classifier to make gesture prediction possible. For allowing the shifting detection as responsive as possible, the selected feature ought to be both simple and fast-processing. To address this problem, we propose a specific gesture based on signal synchronization method [28] that can activate minimum muscular region as muscle contraction.” (section 2.2)

“By means of above electrode shift detection using IPL, the expression of armband rotating position can be confirmed and transformed to Standard Space. Therefore, the shifted degree can be expressed via the offset angle whose channel possessing maximum interpolated amplitude relative to initial electrode configuration.” (section 2.3)

“The above shift detection and correction allow the raw shifted data to transform to unified sEMG one on Standard Space. By means of these proofed data, we can merely train single one classifier without taking random shift conditions into consideration. To evaluate the shift-corrected performance at various sessions over time, we employ above strategy of shift detection and data rearrangement, and set the criterion that whether the corrected position presents under the tolerance of Standard Space (i.e., ±CH 4 or 135°±22.5°).” (section 2.4)

It seems that Formula (2) misses a transposition. Please check it.

Response: Many thanks for this comment. We are sorry for missing this transposition, and this mistake have been corrected. Thanks again.

Figure 3(b) was referenced in the context before Figure 3(a), which might be inappropriate in the order.

Response: The reviewer is right, and we have corrected this issue by swapping their orders and rephasing related sentences.

The discussion section includes content that seems to be experimental results. Please distinguish the content of results and discussion, and list them in the proper section.

Response: We agree with the reviewer, and have adjusted certain content of previous section 4.1 and 4.2 into section 3.2 and 3.3 in revised version. Meanwhile, the remaining content of previous version parts has been rearranged and rewritten to new section 4.1 (titled “Governing parameters varies accuracies”).

Line 195 mentions session 5, which appears to be a small mistake.

Response: Thanks for your kind remark, and we have corrected this mistake with “section 3.2”, please check it.

Some narration is repetitive. For instance, lines 109, 128 and 213 seem to all explain the same thing, that of 8 experiment sessions used for evaluating the prediction. And, in line 213, it says that the 8 independent experimental sessions were used to organize the training and testing datasets. Then, does this mean that the 3 experiment sessions for organizing the ANN training dataset were included in these 8 sessions? If so, is the training data used for testing? This kind of approach is unfair during ANN evaluation.

Response: We appreciate for your careful work and instructive comment. As for repetitive narration, we have removed certain unnecessary content and rephrased some sentences. More importantly, we have not used any part of training data for testing, and truly sorry for our confusing description. Actually, before prediction performance evaluation, sEMG data acquirement from 3 training wearing positions has been conducted independently, we have provided this text at section 2.1.3 in revised manuscript, which means 3 training set and 8 testing set are collected from different sessions and they are independent with each other.

Author Response

GENERAL RESPONSE: We sincerely thank the editor and all reviewers for their valuable feedback that we have used to improve the quality of our manuscript. The reviewer comments are laid out below in italicized font and specific concerns have been numbered. Our response is given in normal font and changes/additions to the manuscript are using the "Track Changes" function in Microsoft Word.

POINT-TO-POINT RESPONSE:

COMMMENTS FROM REFEREE #2:

From the manuscript, it is understood that the authors have carried out rigorous experiments as part of the work and the final accuracy achieved for this work is appreciable. Even then, it is recommended that the below issues regarding the manuscript has to be solved for better representation of the work.:

Response: Thank you for your instructive comments on our article. According to your suggestions, we made several modifications to our previous draft, and the response to the pointed-out issues are as follows.

The manuscript lacks clarity at many places and there are grammatical errors in different sentences. Hence, the usage of English has to be greatly improved in this manuscript. For instance, Page 1: Line 14 and 15, Line 29 and 30, Line 32, 33 and 34; Page 2: Line 52 and 53; Page 4: Line 150 and 151; Page 5: Line 165 and 166; Page 6: Line 180, 181 and 182; and so on…

Response: Thanks for your suggestion and sorry for the poor writing. We have tried our best to polish the language in the revised manuscript. And we also appreciate for your warm work earnestly, and hope that the improvement will meet with approval.

An example is provided below for understanding: Page 1: Line 14 and 15: “Unlike real-time recognition which outputs target gestures waiting until hand motions terminate, our proposed advanced prediction can provide the same results before signals collection finished”. The above sentence can be rewritten as below for better understanding of the sentence: “Unlike real-time recognition which outputs target gestures only after the termination of hand motions, our proposed advanced prediction can provide the same results even before the completion of signal collection.”

Response: Thank you for your instructive comment, we have spared no efforts to rephase these sentences. Many thanks for your careful work.

More details regarding ANN training has to be included in the manuscript. For instance, what regularization method is used in ANN? What about the different layers and neurons involved in ANN?

Response: Thank you for raising this issue. Indeed, the paper lacks detailed information about the architecture of the implemented ANN. Therefore, we have added the following explanation to the Part ANN-Based Hand Gesture Prediction (in section 2.5):

“For outputting predicted gestures before motions finished and decreasing spurious misclassification errors, we add postprocessing to the last link of prediction system. By combining the majority voting method [30], every sliding window (200ms or 40 sampling points) generates one specific code in accordance with predefined gestures via the prediction function of educated classifier. Then, the current window shifts into the next with the step of 5ms. If the gestural labels accumulate to the threshold (55), the postprocessing terminates.

Our forward-propagation ANN model is implemented with three layers and trained this network by using full batch gradient descent with cross-entropy cost function, and applied regularization using weight decay (λ = 750 / total number of vectors). The number of input layer nodes is 376, that is, the product of one sliding window length and total channel number (40 * 8 = 320), plus features extraction functions creating a vector of 56 features (7 * 8 = 56), corresponding to the length of feature vectors. Meanwhile, we find out the optimal size of hidden layer is half of input one using tanh transfer function, the classifier can get best prediction performance in such setting. Moreover, we set 6 nodes on the output layer in order to correspond to the 6 pre-defined gestures (including NM). The specific prediction procedure will be detailed at Algorithm.” (in section 2.5)

The authors have mentioned that they have used 10 subjects to perform pre-defined hand gestures at 8 independent experimental sessions to organize the training and testing datasets. More details regarding the dataset including the training set and testing set used in ANN has to be included in the manuscript.

Response: We appreciate for your careful work and this instructive comment. As you suggested, we have provided more details about how we organize the training and testing set in our revised manuscript. Notably, before prediction performance evaluation, sEMG data acquired from 3 training wearing positions has been conducted independently, which means 3 training set and 8 testing set are acquired from independent sessions. We are truly sorry for our confusing description, this these related contents have also been revised, please check it in section 2.1.3 and section 2.4.

Reviewer 3 Report

The following remarks are all criticisms of your work. Please do not interpret that as a personal attack. I am very impressed with the high recognition accuracy and the fast prediction of your approach. The remarks point out, where I see room for improvement, while I haven't written anything about the aspects of your paper, that are already good.

Remarks about the claimed contributions of your work
(1) I assume, the stated accuracy results are with your fast prediction of the gestures. How are the results, if you process the full 2s of data for each gesture? I expect them to be even better, so I wonder, why you haven't included them in the paper.
(2) You claim, that your approach can detect arbitrary angles for the electrode shift and section 2.2 explains plausibly, how you do that. However, section 2.3 looks like you compensate the electrode shift by simply renumbering the electrodes, which gives you the same coarse angle resolution of 45°, that previous publications achieved, too.
(3) You call your correction of the electrode shift "adaptive". With that term, I expect a successive improvement of the angle correction during normal operation. However section 2.2 looks like, you expect a special gesture (WE) to calibrate the electrode positions. While this approach is quick and easy enough to be used in a practical application, it is neither successive nor is the calibration done during normal operation. Therefore I find your use of the term "adaptive" misleading.

Further remarks are:
- You wrote statements like "Myo armband at 200Hz" multiple times. I assume, 200Hz is the sampling rate of the Myo armband. That isn't completely evident in your text. Also, the Myo armband does not sample all sensors in the same rate. 
- Figure 3(b) does not look like a linear interpolation to me, like it was claimed in section 2.2.
- At the end of section 2.2 you write about a 4th order Butterworth low-pass filter without mentioning its cutoff frequency.
- Why are you using this filter in the first place, when the signal's samples are summed up anyway? (Summing up a complete signal is a low-pass operation, that eliminates all frequencies above 0Hz)
- I suggest, that you colorize the cells in you confusion matrix according to the probability of the confusion. That will make them much easier to read.
- You do not describe how you segment the signals, in order to define, which are the 2s during which a single gesture shall be recognized.
- You give a rough description of the majority vote system in section 2.5 and 4.4. However, you never really explain, how it works. Do you have multiple ANNs or do you classify each instance of the sliding window and take the majority of that? If it is the latter, please elaborate, how wide is a window, by how much is it shifted in each sliding step and how many windows you need at minimum for the majority vote.
- Please list the specs of your ANN. How many layers? How many neurons per layer? Which activation function? Did you use any special topology?

Author Response

GENERAL RESPONSE: We sincerely thank the editor and all reviewers for their valuable feedback that we have used to improve the quality of our manuscript. The reviewer comments are laid out below in italicized font and specific concerns have been numbered. Our response is given in normal font and changes/additions to the manuscript are using the "Track Changes" function in Microsoft Word.

POINT-TO-POINT RESPONSE:

COMMMENTS FROM REFEREE #3:

The following remarks are all criticisms of your work. Please do not interpret that as a personal attack. I am very impressed with the high recognition accuracy and the fast prediction of your approach. The remarks point out, where I see room for improvement, while I haven't written anything about the aspects of your paper, that are already good.

Remarks about the claimed contributions of your work

Response: We appreciate for the time and effort the referee has put into the comments as follows. Meanwhile, we have carefully reviewed the comments, made several modifications to our previous manuscript, and the response to the pointed-out issues are as follows.

I assume, the stated accuracy results are with your fast prediction of the gestures. How are the results, if you process the full 2s of data for each gesture? I expect them to be even better, so I wonder, why you haven't included them in the paper.

Response: We appreciate for the thoughtful comment concerning this issue. However, since our proposed gesture prediction method requires that the waiting time should be less than the entire gesture duration, it seems paradoxical to the main topic prediction rather than recognition if we process the full 2s of data in testing set. Meanwhile, several comparisons from previous literatures are listed on Table 4 (Traditional recognition whose response time are all greater than gesture duration), it is should be noted that the final accuracies varied which closely depend on applied classifiers and electrodes. From given recognition results, our model is capable of outputting target gestures with relative enough accuracy.

You claim, that your approach can detect arbitrary angles for the electrode shift and section 2.2 explains plausibly, how you do that. However, section 2.3 looks like you compensate the electrode shift by simply renumbering the electrodes, which gives you the same coarse angle resolution of 45°, that previous publications achieved, too.

Response: We thank the reviewer for raising this issue. Actually, section 2.2 implies currently wearing positions, and these raw channels are endowed with particular angles by plotting interpolated curve based on every sum of preprocessed data in each sensing channel. Notably, these channel-endowed angles without coarse resolution can reflect real spatial wearing position on the basis of proposed Standard Space and discussion of synchronous gesture selection in section 4.1. Meanwhile, the channel with maximum interpolated value also represents channel 4 (CH4) as the description of section 4.1 with its endowed certain angle. So we merely rearrange the shifted channel to Standard Space via proposed strategy. Importantly, we do not change the angle since the current wearing position is not altered. Therefore, the renumbered channels still possess specific wearing position angles as shown in Figure 5(a) rather than with coarse resolution of 45°. Thanks again for pointing this issue out.

You call your correction of the electrode shift "adaptive". With that term, I expect a successive improvement of the angle correction during normal operation. However, section 2.2 looks like, you expect a special gesture (WE) to calibrate the electrode positions. While this approach is quick and easy enough to be used in a practical application, it is neither successive nor is the calibration done during normal operation. Therefore, I find your use of the term "adaptive" misleading.

Response: We agree with the reviewer that such term “adaptive” might cause misleading. As such, we have changed all the misused words with “adaptive” to “simplified rapid” in our revised manuscript. Thank you, again!

Further remarks are:

You wrote statements like "Myo armband at 200Hz" multiple times. I assume, 200Hz is the sampling rate of the Myo armband. That isn't completely evident in your text. Also, the Myo armband does not sample all sensors in the same rate.

Response: Many thanks for this helpful comment, and indeed, the “200Hz” can only represent Myo armband sampling frequency. Therefore, we have changed all the above-mentioned expression to “Myo armband normally works with a sampling rate of 200Hz”.

Figure 3(b) does not look like a linear interpolation to me, like it was claimed in section 2.2.

Response: We appreciate for your careful work, and are sorry for this mistake. Indeed, the introduced interpolation method is not a liner interpolation but a cubic spline interpolation. We have corrected, and please check it out in section 2.2 and Figure 3. Thanks again for pointing this issue out.

At the end of section 2.2 you write about a 4th order Butterworth low-pass filter without mentioning its cutoff frequency.

Response: Thank you for your comment. We have added the cutoff frequency that “with a cutoff frequency of 5 Hz”, and are sorry for missing this information.

Why are you using this filter in the first place, when the signal's samples are summed up anyway? (Summing up a complete signal is a low-pass operation, that eliminates all frequencies above 0Hz)

Response: We sincerely appreciate for pointing this out. There are two main reasons that we applied this low-pass filter in preprocessing section. First, the Butterworth filter owns fewer design parameters than others, for example, elliptic IIR and Chebyshev filters. The filter selection can effectively reduce the computational cost. Second, with the cutoff frequency of 5Hz, we can reduce the additional noise of the acquired sEMG signals and the device, which can keep more effective information for further processing.

I suggest, that you colorize the cells in your confusion matrix according to the probability of the confusion. That will make them much easier to read.

Response: Thank you for your nice suggestion, and we have re-colorized confusion matrix as you suggested. Meanwhile, we have also removed previous Figure 7 and reorganize their data into Table 2 to make this article more concise.

You do not describe how you segment the signals, in order to define, which are the 2s during which a single gesture shall be recognized.

Response: Thank you for your careful work and instructive comment. Sorry for not further explaining data segmentation in this paper. As you suggested, we have provided the following text to describe how we segment the signals in section 2.1.

“For data preprocessing and segmentation, we implement initial data process on the smoothed signals with mark the region of muscular activity. This practice is achieved by calculating the energy spectrum using short-time Fourier transform (STFT), which can eliminate inactive interval and keep effective information. Then, we employ sliding windows to convert signal data into feature vectors. The window length in this study is 200ms with a step of 5ms, and every sliding window can automatically align with the onset of muscular activity region, and slide to next one along the processing direction.” (in section 2.1)

You give a rough description of the majority vote system in section 2.5 and 4.4. However, you never really explain, how it works. Do you have multiple ANNs or do you classify each instance of the sliding window and take the majority of that? If it is the latter, please elaborate, how wide is a window, by how much is it shifted in each sliding step and how many windows you need at minimum for the majority vote.

Response: We appreciate very much for your thoughtful comment, and are sorry for not addressing further explanation concerning this issue. We have added more descriptions regarding to how the majority voting works that we adopted in section 2.5. Moreover, a forward-propagation ANN is applied at classification including classifier training and testing. So, as you suggested from the latter, we also listed the specific parameters and corresponding values also in section 2.5., which are applied in this study.

We have added the following text in section 2.5 to make the processing procedure more evident:

“By combining the majority voting method, every sliding window (200ms) generates one specific code in accordance with predefined gestures via the prediction function of educated classifier. Then, the current window shifts into the next with the step of 5ms. If the gestural labels accumulate to the threshold (55), the postprocessing terminates.”

Please list the specs of your ANN. How many layers? How many neurons per layer? Which activation function? Did you use any special topology?

Response: Thank you for bring this point to our attention. Indeed, the paper lacks detailed information about the specs of our proposed ANN. As the reviewer suggested, we have added the following explanation to the Part ANN-Based Hand Gesture Prediction (in section 2.5):

“Our forward-propagation ANN model is implemented with three layers and trained this network by using full batch gradient descent with cross-entropy cost function, and applied regularization using weight decay (λ = 750 / total number of vectors). The number of input layer nodes is 376, that is, the product of one sliding window length and total channel number (40 * 8 = 320), plus features extraction functions creating a vector of 56 features (7 * 8 = 56), corresponding to the length of feature vectors. Meanwhile, we find out when the optimal size of hidden layer is half of input one using tanh transfer function, the classifier can get best prediction performance in such setting. Moreover, we set 6 nodes on the output layer in order to correspond to the 6 pre-defined gestures (including NM). The specific prediction procedure will be detailed at Algorithm.” (in section 2.5)

Reviewer 4 Report

The authors present a hand gesture prediction model with an emphasis on estimating electrode shift effect.
They firstly investigate the optimal strategy for correcting electrode rotation shift with arbitrary angle and then realize an improved ANN‐based hand gesture prediction.

The topic is very interesting.

The experimental results validate the proposed approach.

The English readability is good.

For these reasons, the article can be accepted after the minor revision.

The list of possible corrections is below:
- Fig.1 should be clarified.
- It's not clear the consistency of training and testing set.
- Please explain in more details the architecture of the implemented ANN.
- Skin electrodes could have an important impact on the measurements quality. Please provide more details about the used electrodes and armband.

Author Response

GENERAL RESPONSE: We sincerely thank the editor and all reviewers for their valuable feedback that we have used to improve the quality of our manuscript. The reviewer comments are laid out below in italicized font and specific concerns have been numbered. Our response is given in normal font and changes/additions to the manuscript are using the "Track Changes" function in Microsoft Word.

POINT-TO-POINT RESPONSE:

COMMMENTS FROM REFEREE #4:

The authors present a hand gesture prediction model with an emphasis on estimating electrode shift effect. They firstly investigate the optimal strategy for correcting electrode rotation shift with arbitrary angle and then realize an improved ANN‐based hand gesture prediction.

The topic is very interesting. The experimental results validate the proposed approach. The English readability is good. For these reasons, the article can be accepted after the minor revision.

The list of possible corrections is below:

Response: We appreciate the reviewer’s full approval on our manuscript. Below are the point-to-point response to the comments.

Fig.1 should be clarified.

Response: Thank you for your helpful comment, and we agree with the reviewer that Fig.1 should be clarified in a more specific way. As such, we have colorized certain related zones, numbered the signal processing sequence, titled certain parts. Besides, necessary details have also added below this figure. Thank you again.

It's not clear the consistency of training and testing set.

Response: Thank you for this instructive comment. As such, we have provided more details in section 2.1.3 and section 2.4 to improve the consistency of training and testing set, please check. Thanks again, and hope our improvement will meet the approval.

Please explain in more details the architecture of the implemented ANN.

Response: Thank you for instructive comment. Indeed, the paper lacks detailed information about the architecture of the implemented ANN. Therefore, we have added the following explanation to the Part ANN-Based Hand Gesture Prediction (in section 2.5):

“Our forward-propagation ANN model is implemented with three layers and trained this network by using full batch gradient descent with cross-entropy cost function, and applied regularization using weight decay (λ = 750 / total number of vectors). The number of input layer nodes is 376, that is, the product of one sliding window length and total channel number (40 * 8 = 320), plus features extraction functions creating a vector of 56 features (7 * 8 = 56), corresponding to the length of feature vectors. Meanwhile, we find out when the optimal size of hidden layer is half of input one using tanh transfer function, the classifier can get best prediction performance in such setting. Moreover, we set 6 nodes on the output layer in order to correspond to the 6 pre-defined gestures (including NM). The specific prediction procedure will be detailed at Algorithm.” (in section 2.5)

Skin electrodes could have an important impact on the measurement’s quality. Please provide more details about the used electrodes and armband.

Response: We agree with the reviewer that this factor can deeply influence the quality of acquiring signal information. As the reviewer suggested, we have provided more details about the employed electrodes and armband in our revised manuscript in section 2.1.

“All experiments are conducted with 10 able-bodied subjects (2 females included, 24 ± 1.5 years) by Myo armband, a commercially available sEMG sensor, which consists of 8 circular-distributed bipolar dry electrodes. This sensor is capable of normally measuring forearm muscular electrical activity with sampling frequency of 200Hz. Additionally, the collected sEMG data can be transmitted to host computer (OS: Windows 10; CPU: i7-9750H; RAM: 16GB) via built-in Bluetooth in real time, then, with a front panel graphical user interface for data display and via MATLAB for post data processing and analysis.” (in section 2.1)

Round 2

Reviewer 1 Report

The modifications do make the paper more logical.
Small mistakes:
(1) Please check the headers (column names) of Table 3.
(2) line 328 says "This study also provides a new insight into -simplified rapid correction strategy on ...", is the '-' before 'simplified' extra?

Author Response

GENERAL RESPONSE: We sincerely thank the editor and all reviewers for their valuable feedback that we have used to improve the quality of our manuscript. The reviewer comments are laid out below in italicized font and specific concerns have been numbered. Our response is given in normal font and changes/additions to the manuscript are using the "Track Changes" function in Microsoft Word.

POINT-TO-POINT RESPONSE:

COMMMENTS FROM REFEREE #1:

  1. The modifications do make the paper more logical.

Response: We appreciate the reviewer’s full approval on our manuscript. Meanwhile, we have carefully reviewed the comments, made several modifications to our previous manuscript, and the response to the pointed-out issues are as follows.

  1. Please check the headers (column names) of Table 3.

Response: Thank you for your careful work and instructive comment. We have corrected the column names in accordance with the description below the Table 3. Thanks again.

  1. line 328 says "This study also provides a new insight into -simplified rapid correction strategy on ...", is the '-' before 'simplified' extra?

Response: We appreciate very much for your careful work. Indeed, the '-' before 'simplified' is extra, and we have removed this hyphen ('-'), please check it. Thanks again.

Reviewer 2 Report

There are still few issues in sentence formations and grammar in the manuscript. So the authors are encouraged to go through the entire manuscript and rectify theses corrections. 

Also, the authors should make sure the number of nodes used in each layer of ANN is correct. 

Author Response

GENERAL RESPONSE: We sincerely thank the editor and all reviewers for their valuable feedback that we have used to improve the quality of our manuscript. The reviewer comments are laid out below in italicized font and specific concerns have been numbered. Our response is given in normal font and changes/additions to the manuscript are using the "Track Changes" function in Microsoft Word.

POINT-TO-POINT RESPONSE:

COMMMENTS FROM REFEREE #2:

  1. There are still few issues in sentence formations and grammar in the manuscript. So the authors are encouraged to go through the entire manuscript and rectify theses corrections.

Response: Thanks for your helpful suggestion and sorry for the potential sentence and grammar issues. We have gone through this entire manuscript and tried our best to rectify and polish the language in the revised (r2) manuscript. We sincerely appreciate for your warm work, and hope that the improvement will meet with approval.

  1. Also, the authors should make sure the number of nodes used in each layer of ANN is correct.

Response: Thanks for your instructive comment. We have rephased some necessary sentences to make the description of ANN model more accurate. About the model, we referred from our previous work which provide the source codes in GitHub. We also have added the specific reference after this sentence, please check it. Thanks again.
